# The New Pharmacological Chaperones PBXs Increase α-Galactosidase A Activity in Fabry Disease Cellular Models

**DOI:** 10.3390/biom11121856

**Published:** 2021-12-10

**Authors:** Pedro Besada, María Gallardo-Gómez, Tania Pérez-Márquez, Lucía Patiño-Álvarez, Sergio Pantano, Carlos Silva-López, Carmen Terán, Ana Arévalo-Gómez, Aurora Ruz-Zafra, Julián Fernández-Martín, Saida Ortolano

**Affiliations:** 1Departamento de Química Orgánica, Universidade de Vigo, 36310 Vigo, Spain; pbes@uvigo.es (P.B.); carlos.silva@uvigo.es (C.S.-L.); mcteran@uvigo.es (C.T.); 2BIOILS Group, Galicia Sur Health Research Institute (IIS Galicia Sur), SERGAS-UVIGO, 36312 Vigo, Spain; 3Rare Diseases and Pediatric Medicine Group, Galicia Sur Health Research Institute (IIS Galicia Sur), SERGAS-UVIGO, 36312 Vigo, Spain; maria.gallardo@iisgaliciasur.es (M.G.-G.); tania.perez@iisgaliciasur.es (T.P.-M.); lucia.patino@iisgaliciasur.es (L.P.-Á.); jorge.julian.fernanadez.martin@sergas.es (J.F.-M.); 4Laboratory of Biomolecular Simulations, Institut Pasteur de Montevideo, Montevideo 11400, Uruguay; spantano@pasteur.edu.uy; 5Department of Internal Medicine, University Hospital of A Coruña (CHUAC-SERGS), 15006 A Coruña, Spain; ana.arevalo.gomez@sergas.es; 6Department of Internal Medicine, Hospital de la Serranía, Ronda, 29400 Malaga, Spain; dodita85@hotmail.com

**Keywords:** Fabry disease, pharmacological chaperones, *GLA* variants, Migalastat, lysosomal storage diseases

## Abstract

Fabry disease is an X-linked multisystemic disorder caused by the impairment of lysosomal α-Galactosidase A, which leads to the progressive accumulation of glycosphingolipids and to defective lysosomal metabolism. Currently, Fabry disease is treated by enzyme replacement therapy or the orally administrated pharmacological chaperone Migalastat. Both therapeutic strategies present limitations, since enzyme replacement therapy has shown low half-life and bioavailability, while Migalastat is only approved for patients with specific mutations. The aim of this work was to assess the efficacy of PBX galactose analogues to stabilize α-Galactosidase A and therefore evaluate their potential use in Fabry patients with mutations that are not amenable to the treatment with Migalastat. We demonstrated that PBX compounds are safe and effective concerning stabilization of α-Galactosidase A in relevant cellular models of the disease, as assessed by enzymatic activity measurements, molecular modelling, and cell viability assays. This experimental evidence suggests that PBX compounds are promising candidates for the treatment of Fabry disease caused by mutations which affect the folding of α-Galactosidase A, even for *GLA* variants that are not amenable to the treatment with Migalastat.

## 1. Introduction

Fabry Disease (FD, OMIM #301500) is a genetic disorder caused by mutations in *GLA* (NC_000023.1, Xq22) [1], with an incidence estimated from 1:40,000 to 1:117,000. However, this value can reach up to 1:3000, according to newborn screening studies or including late-onset variants [2,3]. *GLA* encodes for α-Galactosidase A (α-GalA, EC3.2.1.22), a rate limiting enzyme in lysosomal metabolism. The enzymatic deficit leads to the progressive accumulation of glycosphingolipids, such as globotriaosylceramide (Gb3) and its deacylated form Lyso-Gb3, within the lysosome of a variety of cell types (endothelial cells, smooth muscle cells, renal podocytes, tubular cells, glomerular endothelial, mesangial and interstitial cells, cardiac cardiomyocytes fibroblasts, and nerve cells) [4]. FD courses as a progressive disorder characterized by painful small fiber neuropathy, cardiac disease, chronic renal insufficiency, and a high predisposition for cerebrovascular strokes [5]⁠. Currently, FD is treated by enzyme replacement therapy (ERT), which consists in biweekly intravenous (i.v.) injections of recombinant human α-GalA (agalsidase alfa or agalsidase beta) or the orally administrated pharmacological chaperone (PC) Migalastat (also known as 1-deoxygalactonojirimycin, DGJ).

ERT has been the gold standard for the treatment since 2000 [6], although it presents important limitations such as the low half-life and bioavailability of the recombinant enzyme, its incapability to cross the blood–brain barrier, and the frequent activation of the native immune system. DGJ, an iminosugar mimicking structure of D-galactose, is orally administrated and overcomes some of the limitations of ERT, since it is widely distributed, can cross the blood–brain barrier, and does not activate the immune system [7].

Clinical trials demonstrated that this drug is generally well tolerated, stabilizes renal function, and significantly reduces the left ventricular mass index. Lyso-Gb3 levels in plasma and Gb3 levels in kidney interstitial capillary are significantly reduced in naïve patients treated with Migalastat, compared to placebo-treated patients. Gastrointestinal symptoms are also meliorated in treated patients [8,9].

DGJ acts like a PC, stabilizing the structure of the mutant enzyme and avoiding its degradation by the quality control mechanisms of the endoplasmic reticulum, which allows the enzyme to traffic to the lysosome. Once the complex has reached the target organelle the PC is released from the enzyme, upon acid hydrolysis, and the protein can catalyze substrate degradation, still in the presence of an amino acid substitution [10].

Nevertheless, this mechanism of action is efficient only when the mutation is located in specific areas within the protein (i.e., in proximity of the active site), and partially affects the folding of the enzyme. Therefore, the use of Migalastat is only indicated for FD patients with amenable mutations in *GLA* ([11] consulted 31 October 2021).

Considering the relatively and increasingly high incidence of the disease, new molecular scaffolds are needed to provide therapeutic alternatives for treatment of this distressing disease. Therefore, the aim of this work was to assess the efficacy of additional galactose analogues, which stabilize α-GalA, and evaluate their potential use as PCs for the treatment of FD in patients with mutations that are not amenable for the treatment with Migalastat.

## 2. Materials and Methods

### 2.1. Synthesis of PBX Molecules

Chemical reagent 1,2:3,4-Di-*O*-isopropylidene-D-galactopyranose was purchased from Acros Organics (part of Thermofisher, Fair Lawn, NJ, USA) All other chemical reagents were obtained from Sigma-Aldrich (part of Merck, Darmstdt, Germany). Solvents were distilled and dried before use according to standard procedures. Reactions were monitored by analytical thin-layer chromatography (TLC) using silica gel precoated plates (Merck 60 F_254_, 0.25 mm, Merck, Darmstdt, Germany) and flash column chromatography was carried out using silica gel Merck 60 (230–400 mesh) (Merck, Darmstdt, Germany). NMR spectra were performed on a Bruker ARX400 spectrometer (Bruker, Fällanden, Swiss); chemical shifts (*δ*) are given in parts per million (ppm) downfield of TMS as internal standard and coupling constants (*J*) are reported in hertz (Hz). High-resolution mass spectra (HRMS) were recorded on a Bruker microTOF-Focus spectrofotometer.

### 2.2. Mutant Plasmids

Mutant plasmids were generated with QuikChange Lightning Site-Directed Mutagenesis Kit (Agilent, Santa Clara, CA, USA) using vector pR-M10-αGalA [12] as wild type template and following the manufacturer’s instructions. Specific primers (Appendix A) were used to introduce selected mutations, which affect protein folding (p.Asp170Val, p.Pro205Ser, p.Gln279Arg, p.Arg301Gln), in *GLA*.

PCR products were used to transform *XL10-Gold* ultracompetent cells (Stratagene, (part of Thermofisher, Fair Lawn, NJ, USA) and mutated plasmids were purified using the DNA extraction kit NZYMiniprep (Nzytech, Lisboa, Portugal) before sequencing by the Sanger method to confirm the presence of the introduced mutations.

Finally, mutated plasmid maxipreps were prepared from transformed *E. Coli*, using the HiSpeed Plasmid Maxi Kit (Qiagen, Hilden, Germany).

### 2.3. Cell Culture and Transfection

The 293T Hek cells (Human Embryonic Kidney 293T cells #CRL-3216, ATCC, Manassas, VA, USA) were cultured in Dulbecco’s Modified Eagle’s medium (DMEM, #31966021, Thermofisher, Fair Lawn, NJ, USA) with Fetal Bovine Serum (FBS, #A4766801 Thermofischer) 10% (*v*/*v*), Penicillin-Streptomycin (#03-031-1C, Biological Industry, part of Thermofischer Fair Lawn, NJ, USA) 1% (*v*/*v*) and Amphotericin B 0.25 μg/mL (#A9598, Sigma-Aldrich, part of Merck, Darmstdt, Germany), at 37 °C and 5% CO_2_. Cells were transfected with Lipofectamine 3000 (#L3000015, Thermofisher), following the manufacturer’s instructions and using 0.75 µg of each plasmid.

The synthesized compound PBXs (PB48, PB49, PB51) were added to the medium at the concentration of 5 or 10 μM and incubated for 72 h at 37 °C and 5% CO_2_. Compounds DGJ, (1-Azido-β-deoxy-D-Galactopyranoside, #D9641), 1-Azido-1-deoxy-β-D-galactopyranoside (ADBGP, #513989); Allyl-α-D-Galactopyranoside (AaBGP, #48149-72-0) and galactose (#G0750) were ordered from Merk (Merck, Darmstdt, Germany). PB48 is also commercially available at Merck (6-Azido- 6-deoxy-D-Galactopyranoside, #712760).

Cells were harvested by scraping in Phosphate Buffered Saline (PBS) medium and lysed by 3 cycles of freeze and thaw, followed by sonication. Cytosolic fractions were collected by centrifugation at 10,000× *g*, 30 min at 4 °C, before measuring α-GalA activity.

### 2.4. FD Patients’ Lymphocytes Treatment

Upon written consent, we obtained blood samples from patients who were previously diagnosed with FD and are followed at Hospital Álvaro Cunqueiro (SERGAS), Vigo, Spain, as well as at other Spanish national centers. The study was approved by the Ethics Committee of Pontevedra-Vigo-Ourense, Galicia Autonomic Government, Spain (code: 2017/569; 21 December 2017). Enrolled patients present mutations in *GLA* (p.Gln279Arg, p.Leu131Gln, p.Met290Thr), which are involved in α-GalA protein folding. Peripheral Blood Mononuclear Cells (PBMCs) were purified from whole blood extracted in Heparin-Na^+^ by Ficoll-Paque-Plus (#GE17-1440-02 Healthcare, Chicago, IL, USA) density gradient, following blood dilution (1:1) with PBS. After centrifugation (35′ at 805× *g* withouth brake, r.t.), white cells were collected, washed twice with PBS and cultured with Roswell Park Memorial Institute medium (RPMI, #27016) with FBS 10% (*v*/*v*), Penicillin-Streptomycin 1% (*v*/*v*) and Amphotericin B 0.25 μg/mL at 37 °C and 5% CO_2_. Cells were seeded in 24 well plates at a concentration of 20,0000 cells/mL and treated with the PBX compounds at either 5 or 10 μM for 72 h. Finally, cells were harvested and lysed with the same protocol used for transfected cells.

### 2.5. α-GalA Activity

Activity of α-GalA was measured using the method of Chamoles et al. [13]. Briefly, cell lysates activity was measured in the presence of the substrate, 4 mM 4-methylumbelliferyl-α-D-galactopyranoside (4-MU-α-Gal, Glycosynth, #44039, Warrington WA2 8QU, UK) and 50 mM N-acetyl-D-galactosamine (#A2795, Sigma-Aldrich, part of Merck, Darmstdt, Germany) in 0.15 M Phosphate-Citrate buffer (pH = 4.2). The reactions were incubated for 2 h at 37 °C and, at this point, the stopping solution (0.1 M ethylenediamine, pH = 11.4) was added to halt the activity. Then, sample fluorescence was measured, using Twinkle LB 970 Fluorometer (Berthold, Baden Württemberg, Germany), at 360 nm excitation and 450 nm emission wavelengths. Hydrolysed μmoles of substrate were calculated with reference to a 4-methylumbelliferone standard curve and divided for incubation time and total amount of protein in the cell lysate. Total protein concentration was measured with Pierce™ BCA Protein Assay Kit (Thermo-Fisher, Thermofisher, Fair Lawn, NJ, USA) following the manufacturer’s instructions.

### 2.6. α-GalA Kinetics

Recombinant human α-GalA (#6146-GH, R&D Systems, Minneapolis, MN, USA) was pre-incubated at either 37 °C, 39 °C, or 42 °C for 15 min with different concentrations of PB48 (150, 75, 50, 20, 10, 5, 2, 0.5, 0 μM) in Phosphate-Citrate buffer 0.15 M (pH = 4.2) with HALT protease inhibitor. After pre-incubation, the substrate 4-MU-α-Gal was added at different concentrations (4096, 3034, 2048, 1024, 512, 256, 128, 64, 32, 16, 0 μM) to each enzyme/compound solution and the product formation was assessed by measuring fluorescence at 37 °C (360/450 nm) after 30 min. The reaction was stopped with 0.1 M ethylenediamine pH = 11.4 and the product was quantified relative to a standard 4-methylumbelliferone curve. Another experiment was carried out pre-incubating the wild type enzyme with PB48 (150, 100, 75, 50, 10, 0 μM) in Phosphate-Citrate buffer 0.15 M at either pH 4.2 or pH 6.5 for 15 min and assessing substrate cleavage at 37 °C after the addition of 4-MU-α-Gal at the concentrations of 4096, 3034, 2048, 1024, 512, 256, 64, 0 μM. Interpolated Michaelis Menten curves of enzymatic reaction velocity versus substrate concentration and its reciprocal (Lineweaver–Burk plots) were obtained using Graph Pad Prism 9.1 (San Diego, CA, USA).

### 2.7. Molecular Modelling

The construction of a p.Asp170Val mutant was performed by using the mutagenesis toolkit in PyMol [14] (http://www.pymol.org/pymol, 20 October 2021) and a series of molecular docking experiments were performed to evaluate the affinity of the native ligand (galactose) and the PC Migalastat, using the Autodock-Vina suite of programs and their associated force field [15,16]. Clustering of the various conformations found for each ligand was performed via default root mean square deviation (RMSD) parameters and an exhaustiveness value of 20 [17]. Relative energy analysis for each cluster and its associated affinity was then evaluated using the force field included in the Autodock-Vina suite.

### 2.8. Compound Cytotoxicity

Cytotoxicity of Hek cells treated with galactose analogues was real-time assessed, using RTCA DP XCelligence (Acea Biosciences, part of Agilent, Santa Clara, CA, USA). Briefly, cells were seeded, at the density of 7500 cells/well, in special plates equipped with electrodes to measure impedance (RTCA E-Palte 16 PET Acea, #00300600890, part of Agilent, Santa Clara, CA, USA) and grown at 37 °C and 5% CO_2_. Following 24 h, cells were treated with increasing concentrations of PB48, PB51, galactose, or DGJ (0 μM; 10 µM; 20 µM; 50 µM; 100 µM; 200 µM; 500 µM; 1 mM) and cell growth was followed up for an additional 72 h. Calculating the variation of impedance on plate surface, which is proportional to cell detachment and death, the RTCA software 2.0 (part of Agilent, Santa Clara, CA, USA) allowed to plot growth curves (cell index versus time) for each treatment and concentration.

### 2.9. Statistics

Normalized α-GalA activity data significance was assessd by ANOVA (one way non-parametric, Kruskal–Wallis test, multiple comparisons), using Graph Pad Prism v9.1 software. IC_50_ (molar) was calculated, using X-Celligence software, from a cell index at each concentration of galactose analogues and plotted versus time. IC_50_ at time 52 h (corresponding to linear growth phase) was extrapolated from these plots to compare compound effects on cell proliferation.

## 3. Results

### 3.1. PBXs Galactose Analogues Synthesis

The 6-substituted galactose analogues PB48, PB49, and PB51 were prepared from 1,2:3,4-Di-*O*-isopropylidene-D-galactopyranose (**1**) as shown in Figure 1, following or adapting previously described procedures [18,19,20].

Synthesis and structural characterization of galactose analogues PB48, PB49, and PB51 are detailed below.

### 3.2. General Procedure for the Preparation of 6-Substituted-6-deoxy-α,β-D-galactopyranosides

The corresponding 6-substituted-1,2:3,4-di-*O*-isopropylidene-D-galactopyranose 4–6 (0.3 mmol) was treated with trifluoroacetic acid/water (4:1, 1 mL). The reaction mixture was stirred at room temperature for 1–4 h and the solvent was evaporated to dryness. The crude solid obtained was washed with H_2_O (3 × 2 mL) and EtOAc (3 × 2 mL). The residue was purified by column chromatography on silica gel (CH_2_Cl_2_/MeOH, 85:15) to afford the desired compound PB48 and PB51, or by reversed-phase Supelclean^TM^ LC-18 SPE tube (H_2_O/MeOH, 9:1) to obtain PB49.

#### 3.2.1. PB48: 6-Azido-6-deoxy-α,β-D-galactopyranoside

PB48 (90% yield) was obtained as a white solid after stirring the reaction mixture for 1 h. PB48 occurred as a 1:1.6 α/β anomeric equilibrium mixture in D_2_O (H-1 integration); R*_f_* = 0.35 (CH_2_Cl_2_/MeOH, 4:1); ^1^H NMR (D_2_O): *δ* = 5.14 (d, *J* = 3.8 Hz, 1H, H-1α), 4.48 (d, *J* = 8.5 Hz, 1H, H-1β), 4.10–4.04 (m, 1H, H-5α), 3.85–3.81 (m, 1H, H-4α), 3.79–3.76 (m, 1H, H-4β), 3.76–3.64 (m, 3H, H-5β, H-3α, H-2α), 3.55–3.50 (m, 1H, H-3β), 3.50–3.31 (m, 5H, H-2β, H-6α, H-6β); ^13^C NMR (D_2_O): *δ* = 96.4 (C1β), 92.3 (C1α), 73.4, 72.6, 71.6, 69.6, 69.0, 68.9, 68.8, 68.1, 50.8 (CH_2_), 50.7 (CH_2_); HRMS (ESI): *m*/*z* [M + Na]^+^ calculated for C_6_H_11_N_3_NaO_5_: 228.05909, found: 228.05963.

#### 3.2.2. PB49: 6-Amino-6-deoxy-α,β-D-galactopyranoside

PB49 (85% yield) was obtained as a brown solid after stirring the reaction mixture for 2 h. PB49 occurred as a 1:1.5 α/β anomeric equilibrium mixture in D_2_O (H-1 integration); ^1^H NMR (D_2_O): *δ* = 5.17 (d, *J* = 3.8 Hz, 1H, H-1α), 4.48 (d, *J* = 7.9 Hz, 1H, H-1β), 4.18–4.12 (m, 1H, H-5α), 3.89–3.85 (m, 1H, H-4α), 3.83–3.72 (m, 3H, H-4β, H-5β, H-3α), 3.72–3.66 (m, 1H, H-2α), 3.57–3.52 (m, 1H, H-3β), 3.42–3.35 (m, 1H, H-2β), 3.21–3.09 (m, 4H, H-6α, H-6β); ^13^C NMR (D_2_O): *δ* = 96.3 (C1β), 92.2 (C1α), 72.4, 71.4, 70.7, 69.7, 69.2, 68.8, 67.9, 66.1, 40.2 (CH_2_), 40.1 (CH_2_); HRMS (ESI): *m*/*z* [M + Na]^+^ calculated for C_6_H_13_NNaO_5_: 202.06859, found: 202.06834.

#### 3.2.3. PB51: 6-Cyano-6-deoxy-α,β-D-galactopyranoside

PB51 (88% yield) was obtained as a white solid after stirring the reaction mixture for 4 h. PB51 occurred as a 1:1.4 α/β anomeric equilibrium mixture in D_2_O (H-1 integration); R*_f_* = 0.36 (CH_2_Cl_2_/MeOH, 4:1); ^1^H NMR (D_2_O): *δ* = 5.12 (d, *J* = 3.8 Hz, 1H, H-1α), 4.48 (d, *J* = 7.8 Hz, 1H, H-1β), 4.28–4.20 (m, 1H, H-5α), 3.92–3.85 (m, 1H, H-5β), 3.83–3.78 (m, 1H, H-4α), 3.77–3.70 (m, 2H, H-3α, H-4β), 3.68–3.61 (m, 1H, H-2α), 3.55–3.49 (m, 1H, H-3β), 3.38–3.30 (m, 1H, H-2β), 2.75–2.64 (m, 4H, H-6α, H-6β); ^13^C NMR (D_2_O): *δ* = 119.0 (CN), 96.4 (C1β), 92.4 (C1α), 72.4, 71.3, 70.2, 70.1, 69.5, 68.8, 67.8, 66.0, 19.4 (CH_2_); HRMS (ESI): *m*/*z* [M + Na]^+^ calculated for C_7_H_11_NNaO_5_: 212.05294, found: 212.05289.

### 3.3. Galactose Analogues PB48 and PB51 Increase α-GalA Activity in Cells Transfected with Wild Type Enzyme or Selected Variants Affecting Protein Folding

Hek cells were transfected with plasmids that encode wild type α-GalA (pGLA) or mutated isoforms of the enzyme (i.e., p.Asp170Val, p.Pro205Ser, p.Gln279Arg, p.Arg301Gln) with affected protein folding. Then, cells were treated with PBX compounds, galactose, or DGJ at concentrations of 5 or 10 µM for 72 h.

Treatment with PB48 increases the activity of α-GalA compared to baseline (non-transfected, non-treated cells) in all the analyzed variants at both tested concentrations (Figure 2A,B). In particular, PB48 increases the activity of α-GalA more efficiently than galactose or DGJ.

PB51 also increased enzyme activity when applied at 5 µM on cells transfected with p.Arg301Gln, p.Gln279Arg, and p.Pro205Ser α-GalA mutants, while it stabilizes the p.Arg301Gln and p.Asp170Val variants at the concentration of 10 µM.

On the other hand, PB49 only partially increases the activity of wild type α-GalA or p.Arg301Gln isoform when applied at the concentration of 10 µM.

Treatment with galactose at the concentration of 5 µM had an inhibitory effect on the activity of p.Gln279Arg, p.Pro205Ser, and p.Asp170Val enzymes. Treatment with DGJ showed stabilizing effect on p.Arg301Gln and p.Pro205Ser variants at the concentration on 10 µM. However, the increment of enzyme activity is lower than the one observed for the treatment with PB48 at the same concentration.

None of the tested compounds determines a significant increase of α-GalA activity, as assessed by Kruskal–Wallis test, probably due to the overexpression of the transfected plasmids and the contribution of the endogenous enzyme to the measured activity values.

Transfection levels were similar in each sample, as assessed by WB (data not shown).

### 3.4. Galactose Analogues with Different Characteristics Compared to PBXs Inhibit α-GalA Activity

We assessed the relation between structure and activity of the compounds at studying by testing the effect of additional galactose analogues. We transfected Hek cells with p.Asp170Val mutant, which affects the interaction of galactose substrate with α-GalA, and therefore allows to understand the differential interactions of the PBX molecules with the active site of the enzyme. We treated the transfected cells with the galactose analogue ADBGP, which presents an N_3_ radical in position 1 of galactose ring, or AaBGP which presents an allyl radical in the same position. Treatment with both compounds at 5 or 10 μM has an inhibitory effect on the p.Asp170Val variant activity. In particular, we found significant differences in normalized α-GalA activity, following treatment with PB48 in comparison with ADBGP at the concentration of 5 µM (Figure 3).

These results show that substrates with different polarity, length of lateral chain, or substituents in different positions, may affect the stabilizing effect of the galactose analogue on α-GalA mutated enzyme.

### 3.5. PB48 Stabilizes α-GalA When Perturbating Temperature or pH of the Enzymatic Reaction

To explain the action mechanism of the new galactose analogues and confirm that PB48 stabilizes α-GalA when it binds to the active site of the enzyme, we followed the kinetics of purified wild type α-GalA at increasing concentrations of the substrate 4-MU-α-Gal (2048, 1024, 512, 256, 128, 64, 16, 0 μM), pre-treating the enzyme with PB48 (50, 20, 10, 5, 2, 0.5, 0 μM) at different temperatures (37–39–42 °C) (Figure 4). When the enzyme is treated with PB48 in the concentration range used for the assays in cellular models, the compound does not significantly change the kinetic of the α-GalA; however, it clearly accelerates enzymatic reactions when pre-incubating the enzyme at high temperatures. This effect is reached for all tested substrate concentrations at the temperature of 42 °C and with substrate concentrations higher than 16 μM at the temperature of 39 °C for each PB48 concentration (Appendix A).

Treating α-GalA with up to 150 μM PB48 (150, 100, 75, 50, 20, 10, 0 μM), the 4-MU-α-Gal cleavage reaction is also accelerated by the treatment with the compound. The velocity of the enzymatic reaction increases, compared to the non-treated control, when the enzyme is pre-incubated at 39 °C and 42 °C (Appendix A) and also when the reaction is assessed at 37 °C at pH 6.5, at least for substrate concentrations higher than 128 μM (Figure 5).

### 3.6. PB48 Establishes Additional Interaction Points within the α-GalA Binding Site Compared to Galactose or DGJ

To evaluate structural implication of the p.Asp170Val, we analyzed the structural changes of α-GalA (PDBid:3S5Z) and its active site upon mutation p.Asp170Ala (PDBid:3TV8). An RMSD fit between both crystallographic structures reveals negligible differences (Appendix A), suggesting that also p.Asp170Val mutation shares the same topology. Docking experiments, using galactose and PB48, on the wild type protein, the variant p.Asp170A, and an in-silico modeled p.Asp170V variant showed comparable binding energies and poses (Appendix A).

Aiming to rationalize the binding differences between galactose, DGJ, and PB48, we compared the docking poses of the different ligands using X-ray structures of acid-alpha-galactosidase A in complex with galactose (PDBid:3GXP) and DGJ (PDBid:3GXT).

PB48 establishes a more extended network of contacts than galactose and DGJ with both wild type protein (Appendix A) and the p.Asp170Val mutant (Appendix A).

Despite the similar binding poses of galactose and PB48 (Figure 6A,B), the Azido group in PB48 adds a salt bridge with Asp93 and makes contact with the indol group of Trp47 (Figure 5C). These highly stabilizing interactions provide a putative explanation for the potent stabilizing effect of PB48. Indeed, similar azido–indol interactions have been reported in other protein–ligand interactions (see, for instance, PDB structures 6HOC and 3O8G). The putative azido–indol interaction at the absolutely conserved Trp47 is relevant because this amino acid is close to the protein–protein interface with the second protomer in the homodimer (Figure 6).

### 3.7. PB48 and PB51 Increase α-GalA Activity in Leukocytes from FD Patients with the p.Gln279Arg Mutation, Which Is Not Amenable to the Treatment with Migalastat

To validate the data obtained in transfected cells, similar studies were carried out in leukocytes extracted from peripheral blood of FD patients treated with PBX analogues.

We cultured cells from healthy subjects (*N* = 2), as well as a hemizygous patient and a heterozygous patient with the p.Gln279Arg variant in presence of PB48, PB51, DGJ (5 or 10 μM), or galactose (5 μM, Figure 7).

Normalized α-GalA activity was increased in all patients’ samples treated with PB48 at the concentration of 5 μM, while it decreases in the hemizygous patient’s cell treated with DGJ, thus confirming that the p.Gln279Arg variant is not amenable for the treatment with Migalastat [11].

To further substantiate results, we assessed the effect of PB48 and PB51 on p.Gln279Arg variant from leukocytes from three different hemizygous patients belonging to the same family. In hemizygous patients’ leukocytes treated with either PB48, PB51, or DGJ (2.5 µM, 5 µM and 10 µM), the activity of α-GalA was significantly increased by treatment with PB48 at the concentrations of 2.5 µM and 5 µM and at higher rates compared to DGJ (Figure 8A). The treatment with PB51 significantly increased normalized α-GalA activity more than DGJ, only when applied at 10 µM (Figure 8B).

### 3.8. PB48 and PB51 Are Also Effective in Cells from FD Patients with the Mutations p.Leu131Gln or p.Met290Thr

PB48 and DGJ were also tested in leukocytes from hemizygous patients with, respectively, the p.Leu131Gln or the Met290Thr variant in *GLA*.

The normalized enzymatic activity of the α-GalA is significantly higher in p.Leu131Gln hemizygous patient cells treated with PB48 at the concentration of 5 µM when compared to untreated cells. It is worth noting that DGJ has no significant effect on this activity (Figure 9A), which in fact is described as not amenable to the treatment [11].

Analyzing samples from a single patient with the p.Met290Thr variant, we found increased normalized α-GalA, only in cells treated with 10 µM PB51 (Figure 9B).

### 3.9. PBX Has No Significant Inhibitory Effect on Cell Growth

Finally, we tested the effect of PBX compounds and galactose on cell growth in the 10 μM to 1 mM range. We monitored real time cell growth by registering impedance of the cells on the plate surface for 72 h after treatment and demonstrated that there are no significant differences in cell growth curves from untreated cells compared to cells treated with PB48, PB51, or galactose (Appendix A). The IC_50_ along the time of the experiment is also comparable among treatments and its value at time 52 h (linear cell growth phase) is, respectively, 8.43 × 10^−5^, 1.4 × 10^−4^, 1.4 × 10^−4^ for galactose PB48 and PB51 (Figure 10). We also assessed cell growth in cells treated with DGJ (0–500 μM range), obtaining comparable results to the PBXs and galactose (IC_50_ = 2 × 10^−5^ at 52 h, Appendix A).

## 4. Discussion

In the present article, we assessed the efficacy of the synthesized galactose analogues in increasing α-GalA activity and acting as possible PCs in the treatment of FD.

We were able to show that α-GalA activity values of wild type and mutated enzymes were increased upon treatment with PB48 or PB51 in different cellular models and experimental conditions (temperature and pH shifts), demonstrating that the enzyme is more stable in presence of PBX analogues.

The effect of PBXs was demonstrated in Hek cells transfected with clinically meaningful variants associated to FD (i.e., p.Arg301Gln; p.Gln279Arg; p.Pro205Ser; p.Asp170Val). This system is a relevant cellular model to show PC activity for α-GalA since a similar assay, carried out in GLP condition, has been approved to evaluate susceptibility of α-GalA variants to the treatment with Migalastat, before considering its administration in FD patients [21]. Our assay was not performed under GLP conditions, but the experimental procedures were very similar to the ones of the approved test.

Moreover, treatment with PB48 of patients’ cells determines a significative increase in α-GalA activity of the p.Gln279Arg or p.Leu103Gln mutated enzymes, which are not amenable to the treatment with Migalastat, while PB51 may stabilize p.Met290Thr enzyme.

As shown by modeling studies, PB48 establishes an interaction with Trp47, which is located at the interface between monomers and, unlike galactose or Migalastat, it can stabilize mutants which are located at a certain distance from the active site (i.e., p.Gln279Arg) because it most likely facilitates monomer interactions and stabilizes the quaternary structure of the protein. Indeed, Try47 is highly conserved in α-GalA and several mutations of this residue have been related to FD classical presentation (i.e., p.Trp47Arg; p.Trp47Gly; p.Trp47Leu; p.Trp47*) [22,23,24].

In addition, molecular analysis of the structures of α-GalA variants found in FD patients, and analyzed in the present study (p.Leu131, p.Gln279Arg, p.Met290Thr), suggested that all of these mutations impair protein folding. In particular, the p.Leu131Gln variant and the Met290Thr variant would place a polar amino acid within a hydrophobic environment. The p.Gln279Arg would place two positive charges one in front of the other, destabilizing the dimerization interface, while the p.Asp170Val, which was studied in transfected Hek cells, would remove a hydrogen bond between the protein and the ligand (Figure 11). PB48 forms an additional salt bridge with Asp93, which further stabilizes the active site and explains the efficacy of PB48 to stabilize mutants where Asp170 residue is affected. On the contrary, both galactose and DGJ need to interact with Asp170 to bind to the active site.

The experimental data obtained upon treatment with galactose analogues confirm that PB48 increases α-GalA activity and therefore suffices to overcome the loss of a hydrogen bond in the p.Asp170Val mutant and stabilizes the protein–protein interface in the case of the p.Gln279Arg.

Thus, the binding and stabilization of different types of interactions highlight the relevance of the azido–indol interaction and the additional salt bridge established by PB48 and suggest that this compound could also be effective against additional mutations that affect protein folding, which were not directly assessed in the present work.

Based on the modeling results, we hypothesized that PBXs can stabilize α-GalA wild type and mutated enzyme structures, thus acting as PC and facilitating translocation of α-GalA to the lysosome for substrate cleavage.

Following the kinetic of 4-MU-α-Gal hydrolysis in presence of PB48 at different temperatures and pH values we were able to confirm that PB48 allows the enzyme to resist environmental changes, therefore confirming that it stabilizes the protein when bound to its active site.

Unlike DGJ, which is a strong α-GalA inhibitor [25], PB48 does not significantly inhibit the enzymatic reaction at the tested concentrations (Appendix A), and therefore it can stabilize the protein without significantly blocking its function.

However, as it binds to the active site, we cannot exclude that PB48 can compete with the substrate when it is administered at higher concentrations than the ones tested in this study (≥75 mM).

Moreover, we have shown that PB48 stabilizes α-GalA at pH 6.5; therefore, it should favour the stabilization of the folding enzyme during its synthesis in the endoplasmic reticulum (≈pH 7). The complex enzyme/PB48 should easily translocate to the lysosome, where PB48 is possibly released due to the very high substrate concentration and the lower pH.

When mutations that affect protein folding are present in the enzyme structure, the stabilizing effect of PB48 should be even higher than the one assessed with the wild type protein, due to the unique interactions that this compound establishes with crucial residues that stabilize dimer formation, and as suggested by docking simulation results (Appendix A).

ERT treatment for FD presents important limitations; therefore, new therapeutic alternatives are definitively needed and some possible strategy is under evaluation in clinical trials [26].

Migalastat (DGJ) is currently the only available oral compound for the treatment of FD, which can overcome some of the ERT limitations (i.e., immunogenicity, bioavailability). However, the use of Migalastat is only allowed in patients with amenable mutations, which are generally variants that surround the active site of α-GalA. In fact, the interactions that this chaperone establishes can stabilize a limited area in proximity of its binding site. Moreover, DGJ has little effect in variants of the active site like the Asp170Val because the chaperoning action of DGJ strictly depends on the interaction of the iminosugar with the carboxylate of the Asp170, as shown by crystallography and other experimental data obtained with α-GalA mutants at the residue 170 [25].

Like Migalastat, PBX compounds still present some limitations, since they can stabilize α-GalA only when the causing mutation does not affect protein synthesis. Therefore, PBXs are willing to be inactive in patients with large deletions, frame-shift mutation, or splicing variants, which may determine null protein production or largely truncated peptides. However, the galactose analogues that we synthesized present the advantage of establishing a wider interaction range with enzyme residues and therefore to better stabilize 3D structure of α-GalA (monomers interactions), even in variants that are currently excluded from the indications of the oral treatment.

There are more than 1000 mutations that have been related to FD (www.hgmd.org, 20 October 2021) and the majority of them (about 70%) are point mutations, most likely related to protein stability defects; therefore, PBXs may represent a new oral option for a considerable number of patients, who present mutations that affect protein folding, including variants that are not amenable for the treatment with DGJ.

PBXs still have to demonstrate their efficacy in vivo in animal models, but given the similarity of the structure of the molecule with galactose and DGJ, it is assumable that pharmacokinetic properties of these compounds will be similar. In vivo studies with DGJ on FD transgenic mouse models showed significant increase of α-GalA activity in heart, kidney, spleen, and liver of transgenic mouse models with mutations affecting enzyme folding (i.e., p.Arg301Gln), as well as decreased levels of Gb3 and Lyso-Gb3 in the affected tissues [27,28,29]. Furthermore, the efficacy of DGJ to stabilize disease progression was proved in clinical trials and open-label extension studies in patients with amenable mutations [30,31].

In addition, we have proven that treatment with PBXs is not cytotoxic since treated Hek cells present growth curves similar to the ones obtained by treatment with galactose or DGJ and comparable or higher IC_50_.

While we still have to confirm whether PBXs are effective in increasing α-GalA activity in vivo, we have to consider that a limited increase in enzyme activity may be sufficient to metabolize accumulated glycosphingolipids and to revert clinical phenotype. Indeed, in several LSDs, substrate accumulation that determines clinical manifestations occurs when residual enzyme activity decays below a certain threshold (about 10% of residual activity or lower), as demonstrated for β-hexosaminidase A and arylsulfatase A [32]. This low threshold is probably justified by the fact that many LSDs are caused by impaired enzymes, which act as catalysts for biological reactions. Additionally, affected cells in most lysosomal disorders can benefit from the cross correction effect [33]. In terms of treatment, these features support that a low improvement in enzyme activity could be highly effective to meliorate clinical signs.

Moreover, PBXs, like other small molecules, may have the advantage to potentiate the effect, the half-life, and the bioavailability of ERT in a possible combinational therapy, since the chaperone is also active on the wild type enzyme, which can also be stabilized.

More efforts also have to be spent on fine tuning the in vivo dose of the drug, since PBXs, binding to the active site of the target enzyme, may act as inhibitor when administered at high doses; however, we showed little or no inhibitory effect of PBXs on α-GalA activity at the tested concentrations. Turnover of the ligand is favored by high substrate concentrations and by binders that present a shorter half-life than the enzyme, as well as reduced binding affinity at lysosomal pH.

On the other hand, the option to inhibit α-GalA activity with PBXs at high concentrations can open new therapeutic possibilities. It is worth considering that lysosomal enzymes participate in a common pathway [34] to cellular metabolism and α-GalA inhibition may help to restore substrate synthesis/cleavage balance in other LSDs or even in other more frequent conditions characterized by lysosomal dysfunction and autophagy blockage.

## 5. Conclusions

The experimental evidence presented in this work suggests that PBX compounds are very promising candidates for the treatment of FD determined by mutations which affect the folding of the protein, even for α-GalA variants that are not amenable to the treatment with Migalastat.

While further preclinical studies are needed to prove in vivo the efficacy of PBXs, we have clearly demonstrated its effectiveness and safety in relevant cellular models of FD, as assessed by enzymatic activity measurements, molecular modelling, and cell viability assays.

## 6. Patents

PBX compounds are protected through the following patents: EP3725310A1; ES2716305A1; ES2716305B2; US2021179575A1; WO2019115854A1.

## Figures and Tables

**Figure 1 biomolecules-11-01856-f001:**
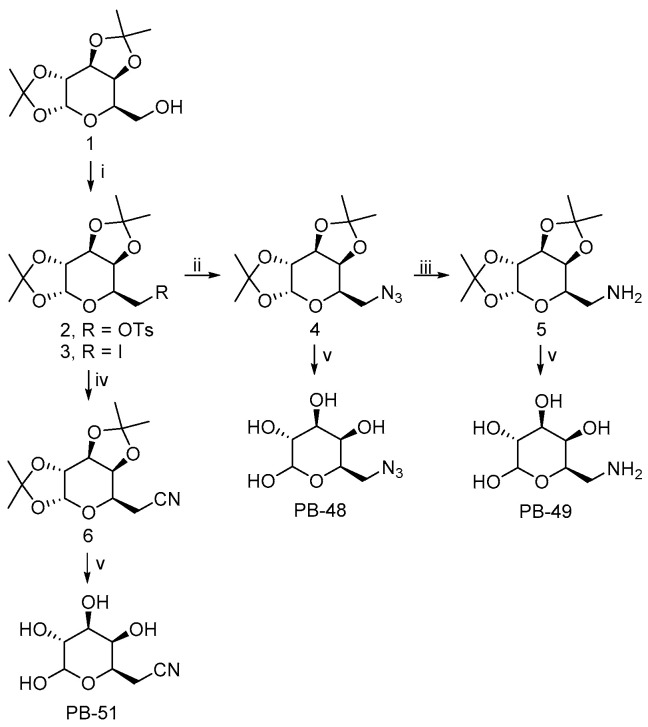
PBX compound synthesis steps. The compounds PB48, PB49, and PB51 are indicated in footnotes. Reagents and conditions: (i) TsCl, pyridine, r.t. 24 h, or I_2_, PPh_3_, imidazole, toluene, CH_3_CN, 80 °C, 4 h, 85% (3); (ii) 2, NaN_3_, DMSO, 115 °C, 24 h, 79% (from 1); (iii) H_2_, Pd/C 10%, EtOH, r.t., 18 h, 99%; (iv) 3, NaCN, DMF, 80 °C, 24 h, 60%; (v) TFA/H_2_O 4:1, r.t., 1–4 h, 90% (PB48), 85% (PB49), 88% (PB51).

**Figure 2 biomolecules-11-01856-f002:**
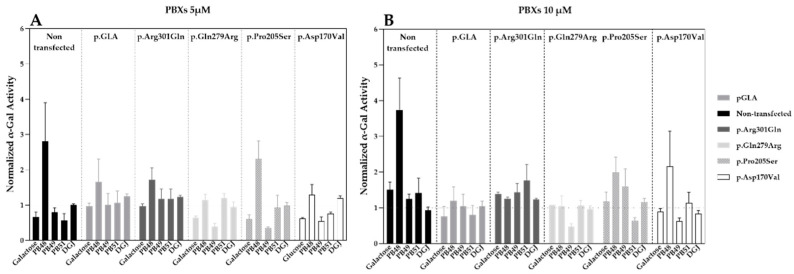
PB48 and PB51 increase the activity of α-GalA in Hek cells transfected with *GLA* variants impairing enzyme folding. Activity of α-GalA assessed in Hek 293T cells transfected with plasmids encoding different α-GalA variants (p.GLA wild type enzyme, p.Arg301Gln, p.Gln279Arg, p.Pro205Ser, and p.Asp170Val) and treated with PBX compounds (PB48, PB49, and PB51), galactose, or DGJ at concentrations of 5 μM (Panel **A**) or 10 μM (Panel **B**). Histograms represent mean normalized activity values for each treatment and *GLA* variant, taking the activity value of non-treated non-transfected control as reference. Error bars represent the Standard Error of the Mean (SEM). Statistical significance was assessed by Kruskal–Wallis test, comparing each condition to non-transfected non-treated reference.

**Figure 3 biomolecules-11-01856-f003:**
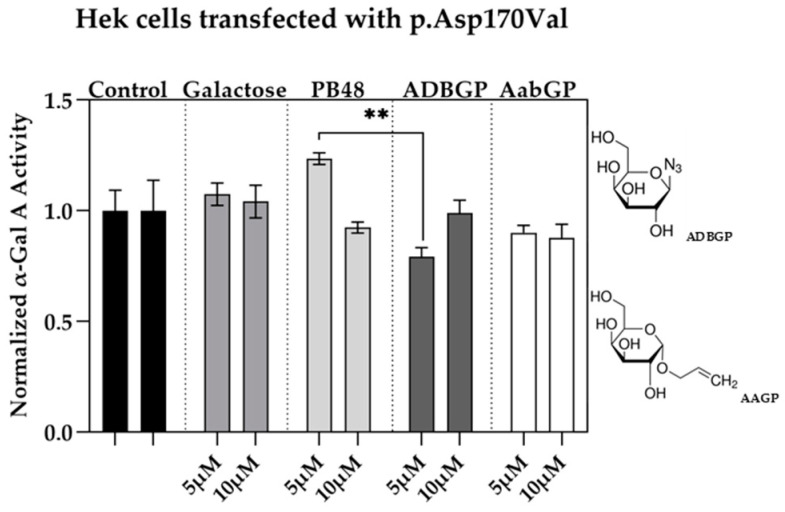
Galactose analogues with substituents in position 1 inhibit α-GalA activity in Hek cells transfected with p.Asp170Val variant. Activity of α-GalA assessed in Hek 293T cells transfected with p.Asp170Val variant, which affects the active site, and treated with ADBGP, AABGP, PB48, or galactose at concentrations of 5 and 10 μM. The histograms represent mean normalized activity values for each treatment, taking non-treated control activity value as reference. Error bars represent ± SEM. Statistical significance was assessed by Kruskal–Wallis test (** *p* ≤ 0.005). The structures concerning galactose analogues ADBGP and AABGP are also represented.

**Figure 4 biomolecules-11-01856-f004:**
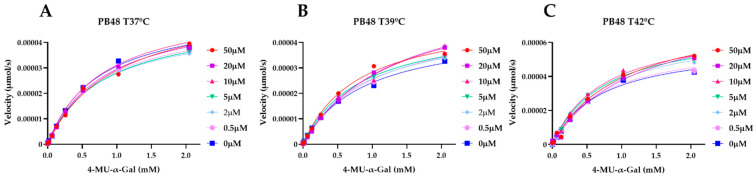
PB48 stabilizes α-GalA when increasing temperature. Interpolated Michaelis–Menten curves with the velocity of 4MU-α-Gal substrate cleavage reaction against substrate concentration. α-GalA was preincubated with PB48 (0–50 μM) at the temperatures of 37 °C (**A**), 39 °C (**B**), and 42 °C (**C**). Error bars represent SEM. The reaction in presence of PB48 at the temperature of 39 °C or 42 °C is faster than the reaction carried out in the absence of the compound.

**Figure 5 biomolecules-11-01856-f005:**
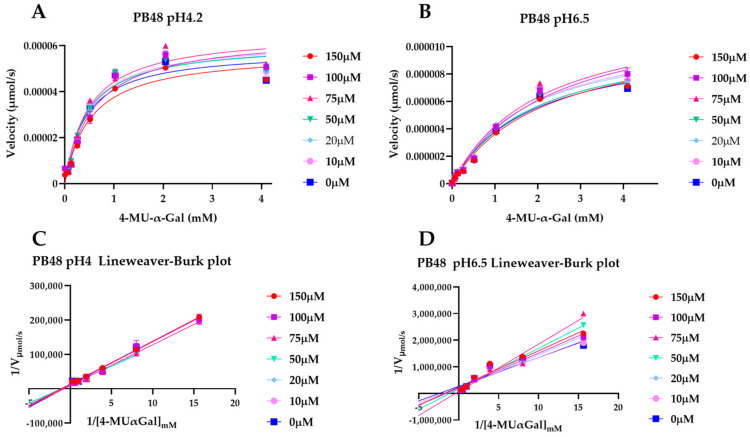
PB48 stabilizes α-GalA at increased pH. Michaelis–Menten interpolated curves of velocity of 4MU-α-Gal substrate cleavage reaction against substrate concentration, following pre-incubation of α-GalA with PB48 (0–150 μM) pH 4.2 (**A**), or pH 6.5 (**B**). Corresponding Lineweaver–Burk plots (**C**,**D**) are also shown. Error bars represents ± SEM. The reaction in the presence of PB48 is accelerated at pH 6.5, compared to the untreated enzyme reaction, at least for substrate concentrations higher than 128 μM.

**Figure 6 biomolecules-11-01856-f006:**
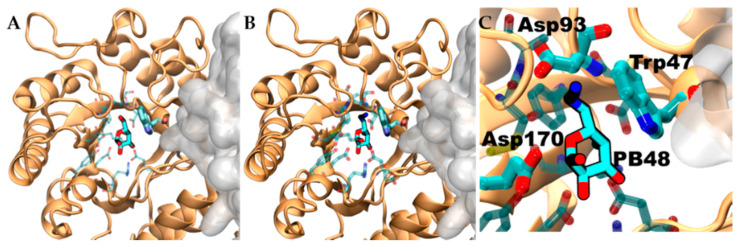
Interaction that PB48 establishes with conserved Trp47 contributes to stabilizing the homodimer formation. Interactions of galactose (**A**) and PB48 (**B**) within the binding site. Ligands are shown in chalky sticks. One protomer in the homodimer is shown in cartoon representation and the nearby chain as a solvent-accessible surface. The amino acids in contact with galactose and PB48 are shown as semitransparent sticks. (**C**) A close up on the PB48 interaction with Asp93 and the conserved Trp47, which is part of the protein–protein interface. Asp170 is also evidenced as a reference.

**Figure 7 biomolecules-11-01856-f007:**
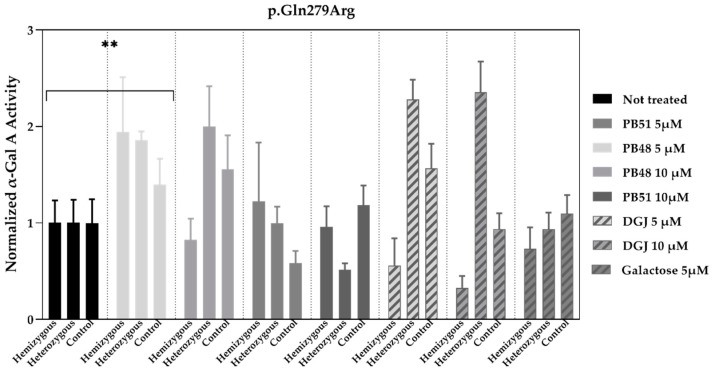
PB48 increases the activity of α-GalA in leukocytes extracted from FD patients with the p.Gln279Arg variant and healthy controls. Activity of α-GalA assessed in leukocytes extracted from FD patients with p.Gln279Arg variant (1 hemizygous and 1 heterozygous) or healthy controls (*N* = 2) and cultured in RPMI with either PB48, PB51, galactose, or DGJ (5 or 10 μM) for 72 h. The histograms represent mean normalized activity values for each treatment and genotype, taking non-treated cell activity value as reference. Error bars represent ± SEM. Statistical significance was assessed by Kruskal–Wallis test (** *p* ≤ 0.005), comparing values obtained in non-treated cells to each treatment and considering the whole group of subjects.

**Figure 8 biomolecules-11-01856-f008:**
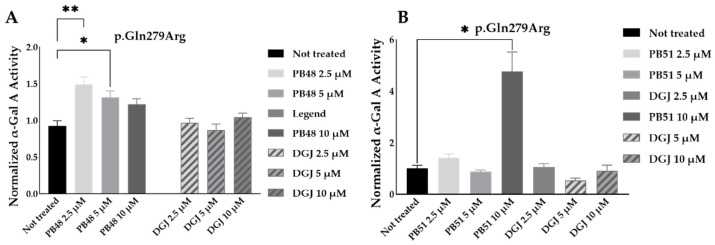
PB48 and PB51 significantly increase α-GalA activity in leukocytes from FD hemizygous patients with the p.Gln297Arg mutation. Activity of α-GalA assessed in leukocytes extracted from FD patients with p.Gln279Arg variant (*N* = 3) cultured in RPMI with either PB48, PB51, or DGJ at 5 or 10 μM for 72 h. (Panel **A**) shows the treatment with PB48 and (Panel **B**) the treatment with PB51. The histograms represent mean normalized activity values for each treatment, taking as reference non-treated cell activity value. Error bars represents ± SEM. Statistical significance was assessed by Kruskal–Wallis test (* *p* ≤ 0.05 and ** *p* ≤ 0.005) comparing each treatment to non-treated control.

**Figure 9 biomolecules-11-01856-f009:**
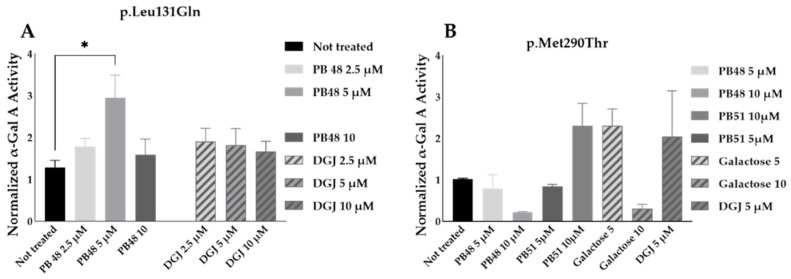
PB48 and PB51, respectively, increase α-GalA activity in leukocytes from FD patients with the p.Leu131Gln and p.Met 290Thr mutations. Activity of α-GalA assessed in leukocytes extracted from an FD hemizygous patient with p.Leu131Gln variant (**A**) and a patient with p.Met290Thr variant (**B**) in *GLA.* Cells were cultured in RPMI with either PBX compounds, galactose, or DGJ at 5 or 10 μM for 72 h. The histograms represent mean normalized activity values for each treatment, taking non-treated cell activity value as reference. Error bars represents ± SEM. Statistical significance was assessed by Kruskal–Wallis test (* *p* ≤ 0.05).

**Figure 10 biomolecules-11-01856-f010:**
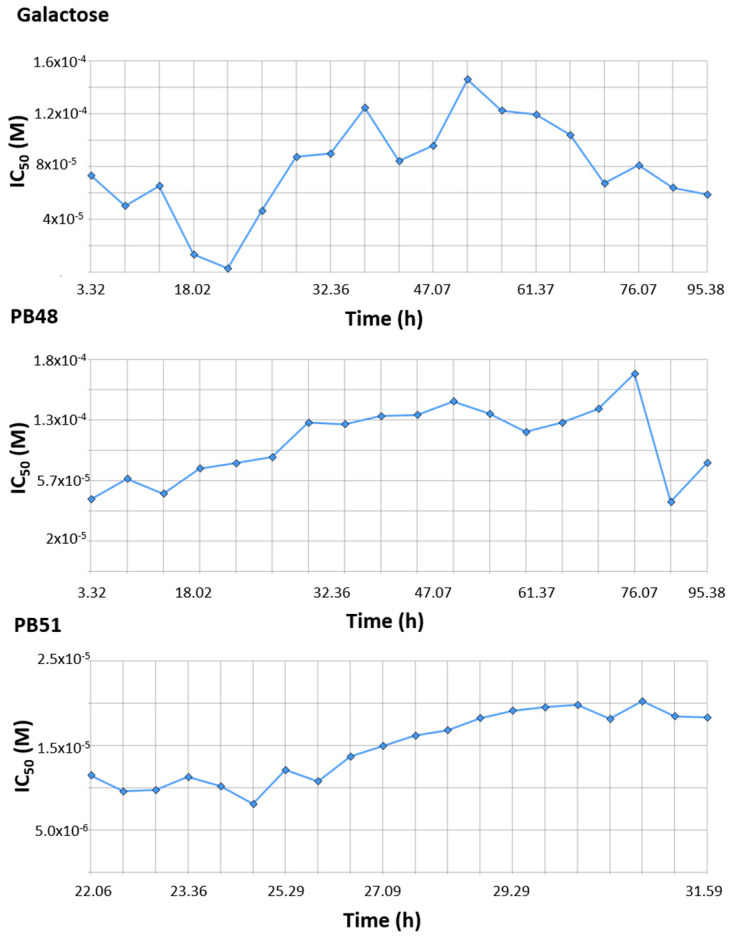
PB48 and PB51 do not affect normal growth of Hek cell. Plot of IC_50_ (M) versus time (hours) for Hek cells treated with PB48, PB51, or galactose (0 μM–1 mM). IC_50_ was derived from cell index versus time curves at each concentration of the corresponding compound.

**Figure 11 biomolecules-11-01856-f011:**
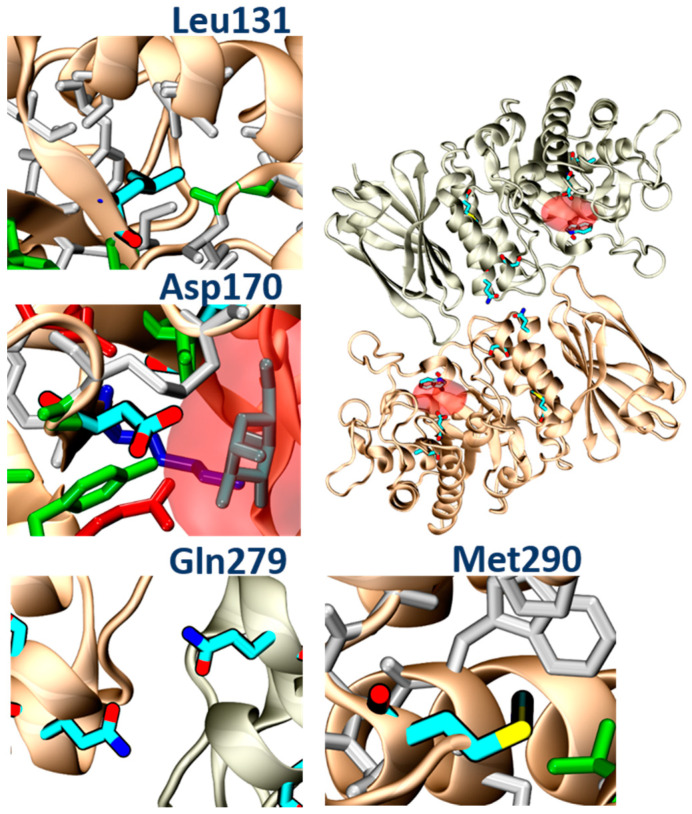
Conformational variations in α-GalA structure determined by mutations that affect protein folding. The upper right panel shows a homodimer with the position of amino acids involved in FD (chalky sticks). A red transparent surface indicates the binding sites. Trp47 is also shown for reference. The side panels show close-ups of the neighborhood of mutating amino acids (Leu131, Asp 170, Gln279, Met290). Neighboring amino acids are color coded—hydrophobic: White; negative: Red; positive: Blue; polar: Green. In the case of Asp170, galactose is also shown in cyan.

## Data Availability

The data presented in this study are available on request from the corresponding author. The data are stored in the public repository of Servizo Galego de Saude (SERGAS, Galician Public Health System), which does not allow external access because to respect the privacy of patients.

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
