# Peer review of "The New Pharmacological Chaperones PBXs Increase α-Galactosidase A Activity in Fabry Disease Cellular Models"

_biomolecules, 2021, doi:10.3390/biom11121856_

Round 1

Reviewer 1 Report

My comments: 

The authors Besada Pedro et alt. in their article „The new pharmacological chaperones PBXs increase α- Galactosidase A activity in Fabry disease cellular models.“ are showing and discussing their idea and model of new possible Fabry disease modifier, or treatment. Since 2000, the enzyme replacement therapy for Fabry disease was introduced. There is growing number of patients being treated in different stage of their lysosomal storage disorder by ERT every other week. There is another type of treatment – migalastat, per oral chaperon treatment, for special group of Fabry patients with amenable variants of GLA gene.  

Analysis and interpretation of the ERT and chaperon treatment efficacies come under scrutiny in last few years. The bioavailability and the half life time, antibody formation on the ERT site and the amenability of GLA variants on the chaperon ´s site gave rise to new drug research. The topic of new possibilities in Fabry disease cohort treatment is relevant and required. The article shows and discuss this theme and describe three galactose analogues in diff cellular models and experimental conditions. From the clinician’s point of view it is very important part of the Fabry disease research, specially experiments with variants associated with AFD related to protein, enzyme stability defects with no amenability by migalastat chaperon treatment.  

The text is easy to read only English language and style are fine/minor spell check required e.g. p.3 101 - Anphotericin B, 102 trnsfected, 125 Anphotericin again... etc. .   

I do not specialize in the methodology & experiments part, I cannot comment this part.  

Author Response

We thank the reviewer for the positive comments on the clinical value of our manuscript, as well as for the opportunity to revise the style and spelling of the manuscript. All the changes in the manuscript were highlighted in yellow.

Reviewer 2 Report

This manuscript describes the development and characterization of certain low molecular weight compounds that conceivably may act as pharmocological chaperones that bind to and stabilize the structure and acti vity of alpha galactosidase A, whose impariment is linked to Fabry disease.  Three compounds are developed and tested, one of which shows a positive effect. Modelling experiments are performed to attain a structural explanation for this effect.

Although the study postulates that the newly developed compounds bind to and stabilize the structure of alpha-galactosidase A, I saw very little actual experimental evidence to support this idea of structural stabilization. A number of rather easily performed experiments are absent, for instance, binding affinity titration analysis by monitoring the activity  to estimate the Kd of each compound, or enzymatic stability assays where the residual activity of alpha galactosidase A is measured after a structural perturbation such as pH shift or temp shift is provided in the presence or absence of compound. Aside from the Pymol simulations, which are ultimately simulations that provide potential binding mechanisms, this study is conspicuously lacking in actual data that point toward a chaperone-like role for these compounds. Such experiments should be added.

Other points; The manuscript is ridden with minor and trivial spelling mistakes.  The test should be checked more carefully.

Author Response

We thank the reviewer for the recommendations and the opportunity to strengthen the value of our study with the suggested experiments. These constructive comments allowed us to improve the quality of the manuscript.

In the first version of the manuscript, we based the proposed mechanism of action for PBXs (pharmacological chaperons) on docking simulations. We agree with the reviewer that this hypothesis should be supported with more experimental data, in spite of the clear effect shown on patient’s cellular models.

In the revised version of the manuscript we clearly demonstrate that PB48, our best candidate, is a competitive inhibitor of α-galactosidase A, which has a lower Ki (about 800 folds) compared to galactose (Fig.3).

We also show that the affinity of the substrate 4-methylumbelliferyl-α-D-galactopyranoside for α-GalA, in presence of the inhibitor PB48, is decreased at higher extent when perturbating pH of the enzymatic reaction. Therefore, we indirectly demonstrated that the enzyme preferentially binds to PB48 at higher pH, because the PB48-enzyme complex is more stable than the complex substrate-enzyme. Unfortunately, we do not dispose of the proper instruments to measure direct binding of the PBXs ligands to the enzyme (filters to measure Tryptophan fluorescence or instruments to assess circular dichroism) and we hope that the new experiments would meet with the reviewer requirements.

We modified the text including the methods, results and discussion of these new data.

(Pag.4, lines:152-167; Pag.9, lines 313-327; Pag.10, Fig.3 and Fig.4; Pag.14, lines 464-467; and Pag.16, lines 516-533). All changes have been highlighted in yellow.

The English style of the article was also thoroughly revised.

Round 2

Reviewer 2 Report

The authors have responded to my review of their original manuscript with a modified version that includes various kinetic analyses of their chemical compound with regard to the enzymatic activity of alpha galactosidase A. They also assert that the manuscript has been re-searched for errors in English.

First of all, my impression of the authors' additional kinetic data is that first, the data is rather insufficient in proving that PB48 is in fact acting as a competitive inhibitor of the enzymatic substrate 4-MU-α-Gal.  The concentration range of  both PB48 and 4-MU-α-Gal seem to be insufficient to evaluate inhibition (compare the high concentration traces for Figure 3 A and B , for example to form a notion on the differences in inhibitory effect), and , since we are not shown the actual analysis of the inhibition (which should be rather easy to show if a Lineweaver-Burk plot at each inhibitor concentration is plotted as a graph), the authors' assertions seem rather weak. Also, I note that the authors have performed analogous experiments in  Figure 4 at two separate pH, where we may directly observe any effects on KM, by comparing the 4-MU-α-Gal concentration at the point of 50% Velocity in each curve; no changes are apparent here, which leads me to suspect that the inhibitory effect is weak indeed, if any.

 On a more fundamental level, changes in the KM of an enzyme at different pH is not an indicator of an inhibitory compound exhibiting a chaperon effect (a protein stabilizing effect) per se. Any number of explanations may be found for the reason why α-GalA shows an increased KM at pH 6.5 compared to pH 4.2, the most prominent among them a change in the charged state of active site residues in the enzyme itself (histidine residues are a major candidate in this hypotheseis). comparison of KM, I am afraid, will not lead to an evaluation of enzyme stability.

 I am curious as to why the authors refuse to perform, perhaps, heat stability experiments where the enzyme is first incubated in the presence of PB48, and the resultant enzyme sample is then assayed for residual enzymatic activity and compared to samples that undergo an identical heat treatment in the absence of PB48. The residual activity is a measure of the amount of enzyme that survived the heat treatment, and if PB48 is indeed stabilizing α-GalA, this value should increase at higher temperatures compared to samples without PB48.  The assay would only require a heat bath in addition to the instruments currently used by the authors, and would be a conclusive evaluation of the role of PB48 as a compound that stabilizes the structure of α-GalA.

My recommendations to the authors; If the enzyme kinetics experiments are going to be used, I would like to see the analyses themselves, not only the raw traces at low 4MU-α-Gal concentrations, or the final elucidated values. The curves should resemble the curves shown in Figure 4, not Figure 3, and derivative plots such as Lineweaver-Burk plots would be preferential. Consider alternatively a heat stability assay such as the one outlined above, as in its present state the data presented seems to argue against an inhibitory role for PB48 (incidentially, you would not want this compound to exhibit a competitive inhibitor effect, I suspect, as the effect would suppress  α-GalA activity in cells, not stabilize it).

Regarding the English and other notations used in the modified manuscript, the grammar is sufficient, but I was more disturbed by the following examples which still remain in the revised version:

Line 91  "QuickChange"          "QuikChange" (it is a product name)

Line 254 to 288 and Figure 1;  also the legends to Figure 2, line 393, line 438: The term "αGalA" and "αGlaA" seem to be used interchangeably, do they denote different things? If so, a different abbreviation would be less confusing.

Line 316 "kinetic"  " kinetics"

line 479 and 483; "Try47" "Trp47"

section 3.4; "Km"  "KM" (please use correct notation)

I repeat again, my search for such errors are by no means extensive. Please go over the manuscript again for similar examples.

Author Response

We thank the reviewer for his critical comments, which highlighted the weak points of the present study and helped us to design more appropriate experiments to show the stabilizing effect of PB48 on α-GalA.

We mostly agree with the referee’s comments and we performed more experiments to amend the incomplete data. We just want to point out that in any moment we were willing to hide or manipulate information, neither we refused to carry out any experiment. The journal granted us very limited time to answer to reviewers (10 days including festivities) and we only had the chance to perform a few experiments (with available reagents and equipments), which resulted to be insufficient and misleading to show the complete kinetics of the enzyme for different substrate and PB48 concentrations. This is the only reason why the curves reported in figure 3 of the first revision did not reach saturation values. We did not intend to hide any data, although we probably carried out a non-accurate analysis based on the previously acquired data. We apologize if assessing a short range of concentrations led us to miss-interpreted conclusions and we hope that the reviewer can consider the new set of data reported in this second revision.  Indeed, as correctly deduced by the reviewer, in the new set of experiments we were not able to calculate a reproducible value of Ki at the tested PBXs concentrations (up to 75mM, Supplementary Fig.6), showing weak inhibitory capacity for PB48. Therefore, we modified this conclusion in the new version of the manuscript.

Moreover, as suggested by the reviewer we now present the experimental data showing Michaelis Menten interpolated curves and Lineweaver plots rather than raw data in all kinetics experiments.

We agree with the reviewer when he suggests that a possible inhibitory effect of PB48 would not be in favor of its use as chaperon therapy. However, we wished to be as transparent as possible in our data report and we assessed the potential inhibitory effect of PB48, as it binds to the active site of the enzyme and it is probable that it can actually compete with the substrate at some concentration. Indeed, the currently approved chaperon for FD, which also binds to the active site, is a potent inhibitor of α-GalA and its dose has to be finely tuned for chaperoning action (Guce et al. Chem Biol. 2011;18(12):1–12., Ref. 25, Ki in the nM range).

Lack of time is also the only reason why we did not include in our first revision the kinetic experiments at different temperatures. In the first round, the reviewer recommended us to show “experiment where the residual activity of alpha galactosidase A is measured after a structural perturbation such as pH shift or temp shift”. We choose to perturbate enzyme through pH shift because we though that this could help us to demonstrate whether PB48 can stabilize the enzyme during its synthesis in the endoplasmic reticulum where the pH is higher than in the lysosome (pH4, optimal for enzymatic reaction).

In the present version of the manuscript,  we show both temperature and pH shifts. We included experiments showing the kinetics of 4-MU-α-GalA substrate cleavage reaction in a wider range of PB48 (0-75mM) and substrate (0-4mM) concentrations. We assessed these reactions at 37ºC, 39ºC or 42ºC and also at pH 4.2 or 6.5 (Fig.3, Fig.4, Supplementary Figures 1,2,6).

We also agree with the reviewer when he/she says that KM is not a direct indicator of enzyme stability and we did not mean to say that KM at pH 6.5 was increasing just because of the effect of PB48. As the referee pointed out, at non-optimal pH the enzyme conformation changes and therefore the affinity for its substrate is also modified. We probably were not clear enough in our explanation, we just wanted to suggest that the variation of the KM at increasing PB48 concentrations was more pronounced at pH 6.5 compared with the KM variation that we observed at pH 4.2 for different PB48 concentrations. In the present version of the manuscript, we removed this affirmation not to generate confusion and we remit to observe the Lineweaver-Burk plots.

Finally, we carefully revised the manuscript again to correct typing errors in the whole text and not limited to the ones specified by the reviewer, which were highlighted in light blue.

In summary, we replaced figure 3 and 4 with new images summarizing the new experiments. We also modified Materials and Methods section (Pag.4, lines 151-165); Results section (Pag. 9-10, lines 311-346) and Discussion section (Pag.14-15, lines 458-464; Pag.16-17, lines 510-530), as well as Supplementary information (included Supplementary Figure 1 and 2). All the modifications were highlighted in blue in the revised version of the manuscript.